# Core Microbiome and Microbial Community Structure in Coralloid Roots of *Cycas* in Ex Situ Collection of Kunming Botanical Garden in China

**DOI:** 10.3390/microorganisms11092144

**Published:** 2023-08-24

**Authors:** Zhaochun Wang, Jian Liu, Haiyan Xu, Jiating Liu, Zhiwei Zhao, Xun Gong

**Affiliations:** 1State Key Laboratory for Conservation and Utilization of Bio-Resources in Yunnan, Yunnan University, Kunming 650091, China; wangzhaochun@mail.kib.ac.cn; 2Key Laboratory of Economic Plants and Biotechnology, Kunming Institute of Botany, Chinese Academy of Sciences, Kunming 650201, China; liujian@mail.kib.ac.cn (J.L.); liujiating@mail.kib.ac.cn (J.L.); 3University of Chinese Academy of Sciences, Beijing 101408, China; 4State Key Laboratory of Plant Environmental Resilience, College of Biological Sciences, China Agricultural University, Beijing 100193, China; haiyan89xz@sina.cn

**Keywords:** *Cycas*, coralloid root, core microbiome, *Desmonostoc*, *Exophiala*

## Abstract

Endophytes are essential in plant succession and evolution, and essential for stress resistance. Coralloid root is a unique root structure found in cycads that has played a role in resisting adverse environments, yet the core taxa and microbial community of different *Cycas* species have not been thoroughly investigated. Using amplicon sequencing, we successfully elucidated the microbiomes present in coralloid roots of 10 *Cycas* species, representing all four sections of *Cycas* in China. We found that the endophytic bacteria in coralloid roots, i.e., *Cyanobacteria*, were mainly composed of *Desmonostoc*_PCC-7422, *Nostoc*_PCC-73102 and *unclassified_f__Nostocaceae*. Additionally, the Ascomycota fungi of *Exophiala*, *Paraboeremia*, *Leptobacillium*, *Fusarium*, *Alternaria*, and *Diaporthe* were identified as the core fungi taxa. The Ascomycota fungi of Nectriaceae, Herpotrichiellaceae, Cordycipitaceae, Helotiaceae, Diaporthaceae, Didymellaceae, Clavicipitaceae and Pleosporaceae were identified as the core family taxa in coralloid roots of four sections. High abundance but low diversity of bacterial community was detected in the coralloid roots, but no significant difference among species. The fungal community exhibited much higher complexity compared to bacteria, and diversity was noted among different species or sections. These core taxa, which were a subset of the microbiome that frequently occurred in all, or most, individuals of *Cycas* species, represent targets for the development of *Cycas* conservation.

## 1. Introduction

Plants do not grow as axiomatic organisms in nature, but rather harbor a diverse community of microorganisms, known as the plant microbiota [1]. Endophytes are a category of plant-recruited microorganisms that can form symbiotic relationships with host plants, originate in seed or rhizosphere microbial communities [2], and colonize different plant organs, including roots, stems, leaves, flowers, fruits, seeds, ovules, and pollens [3,4,5,6]. Colonization of endophytes in plants enables them to resist various stresses, regulate the plant’s immune system, prime the host plant’s defensive response, and improve the plant’s metabolic and developmental status during germination [2,7,8]. Recent studies suggest that plants should be viewed not as separate entities, but rather as complex living organisms resulting from intricate interactions between plant hosts, their associated microorganisms, and the surrounding environment [9,10,11,12,13,14]. The composition of microorganisms in plants is complex, and therefore synergistic or competitive interactions among microorganisms can occur within these communities [9,10]. The assembling of microbial communities are influenced by factors such as host genotype, plant niche, and growing environment [10,11,12]. A balanced microbiome is important for regulating relevant traits in plants, and an imbalance in environmental nutrients could in turn affect the benefits conferred by the microbiome on plants [13,14].

Cycads represent one of the most ancient seed plants, and their relatively primitive morphological characteristics and physiological structures indicate that they are valuable materials for studying the origin and evolution of seed plants [15]. Cycadaceae contains only one genus (*Cycas* L.) with about 120 species, and is the most diversified and extensively distributed group in extant cycads [16]. The *Cycas* genus can be classified into six sections based on their reproductive characteristics, as per the most universally accepted classification systems [17]. However, ca. 68% of cycads are threatened with extinction [18], and all *Cycas* species were listed as first-grade protected plants in China. Notably, cycads display a highly specialized root structure that accommodates an endosymbiotic microbiome and grow to a depth of 30 cm below the ground surface [19,20] (Figure 1). These specific roots repeatedly branch into short and thick branches, forming lumpy nodules called coralloid roots because of their coral-like structure [21] (Figure 1b–d). In the coralloid roots of cycads, a region filled with blue mucous is visible to the naked eye within a ring 1 to 3 mm thick, and electron microscopy revealed that commensal Cyanobacterial cells are localized in the intercellular space of coralloid roots’ cortex parenchyma [22,23] (Figure 1c,d).

The Cyanobacterial cells in the coralloid roots of cycads consist of a group of photosynthetic bacteria that can form symbiotic relationships with a wide variety of hosts in diverse environments [24]. These bacteria are typically found within host organs or tissues in microaerobic environments, and can form symbiotic associations with a wide range of other organisms except cycads, including fungi, lichens, hornworts and liverworts, *Azolla* in ferns, and *Gunnera* in angiosperms [20]. Among symbionts, cycads are the only gymnosperms known to co-exist with *Cyanobacteria*. The predominant symbiotic *Cyanobacteria* detected in cycad coralloid roots are mainly reported as *Nostoc* [25,26,27,28], *Calothrix* [28,29], *Anabaena* [30], and *Desmonostoc* [31].

The ability to establish symbiotic relationships with various groups of microorganisms in coralloid roots has been suggested to be evolutionarily conservative in cycads, possibly to retain the characteristics of adapting to drought and resisting environmental threats [32], but this hypothesis has not been thoroughly examined. Recent studies on endophytic communities of *Cycas* coralloid roots implied significant differences in endophytic bacterial communities between coralloid roots and regular roots in *Cycas*, but no significant difference were observed in endophytic fungal communities [33]. Furthermore, coralloid roots in *Cycas* have specific core microbial taxa, and the composition of microbial community is more affected by small niche differentiation than by geographical environment [3]. The species richness and *Alpha* (α) diversity of fungal community in coralloid roots were significantly higher in *Cycas* collected from common gardens than those from natural habitats [31]. Although the above findings have expanded our knowledge of the diversity, variability, and complexity of the microbiome in cycads, these studies have mainly focused on a single or a pair of species, thus limiting our understanding of the evolution of complex microbial community structures in the coralloid roots of different cycad species.

Despite the complexity of the microbiome in plants, they still harbor a subset of core taxa with which they persistently correlate across diverse environmental gradients. In the context of plant–microbe interactions, core taxa play a pivotal role in facilitating nutrient acquisition, maintaining plant health and productivity, and conferring tolerance to abiotic stress [34,35,36].

Determining the core taxa associated with different taxonomic ranks (i.e., sections or species) may further help to protect *Cycas* resources and contribute to the development of biological control strategies. Here, in the same ex situ conservation environment, we conducted experimental studies to investigate the endophytic microbiome abundance of mature coralloid roots from different *Cycas* species, aiming to (1) characterize the microbial communities associated with different *Cycas* species and (2) identify a ‘candidate core’ community taxa that are globally shared by *Cycas*. By defining the common core microbiome, we discussed their potential roles in the symbiotic assemblies.

## 2. Materials and Methods

### 2.1. Sample Collection and Preparation

Healthy and mature coralloid roots from 34 samples of 10 *Cycas* species were collected from the Kunming Botanical Garden (25°8′14″ N, 102°44′34″ E, Alt. 1885 m) in Yunnan province, China (Table 1). Currently, the distribution of *Cycas* in China is primarily classified into four distinct sections. To enhance our comprehensive understanding of the existing endophytes in *Cycas* coralloid roots within China, core microbiome suitable for most *Cycas* sections were selected whenever possible, and representative species from each of the four sections were chosen. These selected *Cycas* species were all located within the ex situ conservation area of Kunming Botanical Garden. The rationale behind selecting these representative species from each section was as follows: *Cycas panzhihuaensis* (PZH) is the only species in sect. *Panzhihuaenses* (D. Yue Wang) K.D. Hill (PZH), *Cycas revoluta* (ST) is the only species in sect. *Asiorientales* Schuster (AS) [37], *Cycas pectinata* (BC) is the most widely distributed species in sect. *Indosinenses* Schuster (IS), and other species from the sect. *Stangerioides* Smitinand (SS) cover most of the habitats of this section [17]. The design and methodology of the experiment are illustrated in Figure 2. Samples were collected using a sterile sampling bag and stored on ice, then transferred to −80 °C until DNA extraction. Prior to extraction, all samples were surface sterilized according to the previous sterilization pre-experiment. We immersed the samples in 75% ethanol for 30 s, followed by a rinse with sterilized distilled water. Next, we immersed the samples in 2.5% NaClO for 4 min and rinsed them five times with sterile distilled water. After the final rinsing with water, last rinsed water were spread on PDA or beef extract peptone mediums and incubated in the dark at 28 °C for three or four days. No microbial growth was observed on the surface of the media, indicating the effectiveness of the surface sterilization.

### 2.2. Total DNA Extraction, PCR Amplification and Sequencing

DNA was extracted following the cetyltrimethyl ammonium bromide (CTAB) method [38]. Bacterial 16S rRNA genes were selected for amplification using the 515F and 907R [39] primer pair and the internal transcribed spacer (ITS) region for amplification of fungi community using the 1737F and 2043R primer pair [33]. PCR reactions for bacteria and fungi contained 10 ng of template DNA, 0.4 μL of BSA, 0.4 μL of FastPfu polymerase, 0.8 μL of reverse primer (5 μM), 0.8 μL of forward primer (5 μM), 2 μL of dNTP (2.5 mM), and 4 μL of 5X FastPfu Buffer. This reaction was made up to a total volume of 20 μL with molecular biology-grade water. The thermocycling conditions were as follows: 95 °C for 3 min, followed by cycles of 95 °C for 30 s, 55 °C for 30 s, 72 °C for 45 s and a final extension of 72 °C for 10 min. A total of 27 and 35 cycles were used in bacterial and fungal amplification, respectively. The PCR products were quantified using the QuantiFluor™-ST Blue fluorescence quantification system (Promega, Madison, WI, USA), and then mixed proportionally according to the sequencing volume required for samples. The TruSeq^TM^ DNA Sample Prep Kit was used to construct Miseq libraries, while DNA sequencing was conducted on the Illumina MiSeq PE300 platform at Shanghai Majorbio Bio-pharm Technology Co., Ltd. (Shanghai, China) to generate 300 bp paired-end raw reads.

### 2.3. Sequence Processing and Data Analysis

Raw data obtained from Illumina MiSeq sequencing quality were filtered using the Fastp software (v0.19.6) [40]. Forward and reverse sequences were conducted using the FLASH (v1.2.7) software [41] to form contiguous reads. Clean reads were analyzed using QIIME2 pipeline (v.2020.2) [42]. Data denoising reduction processing was performed using Dada2 [43]. Clean sequence reads were assigned to amplicon sequence variants (ASVs). ASVs were mapped to the SILVA 138 database [44] and unite8.0 database [45] in QIIME2, and those classified as chloroplasts, mitochondria were removed from the ASV table.

We used the *rarecurve* function implemented in the R package *Vegan* (v.2.5-7) to generate rarefaction curves. The α diversity indices (Good’s coverage, Shannon and Chao1 index) were also calculated in *Vegan* [46]. Good’s coverage reflected the depth of sequencing, the Chao1 index primarily estimated the number of species, and Shannon diversity measured microbial community diversity. The *Duncan. test* in the R package *Agricolae* was used to detect significant differences (*p* < 0.05) [47]. The *diversity* function in *Vegan* was used to generate standardization of ASV tables [46]. Principal coordinate analysis (PCoA) between sections was performed based on Bray–Curtis similarity matrices using the *Vegdist* function in *Vegan* and rarefied ASV tables. The *adonis*2 function of Permutational Multivariate Analysis of Variance (PERMANOVA) in the *Vegan* package was used to test the difference significance (*p* < 0.05) of the data, and the *betadisper* function was used to evaluate the multivariate homogeneity of groups dispersions (dispersions *p* value > 0.05). Multiple comparison was conducted using the *pairwiseAdonis* (v.0.4) package to further discover which of the two groups were significantly different or not [48]. The complexity and interaction of bacterial and fungal communities were investigated using a co-occurrence network using the R package *psych* (v.2.1.9) based on Spearman correlations and the rarefied ASV tables [49]. Only correlations with r > 0.6 (and *p* < 0.05) were considered to indicate a valid interactive event. The network showed the closeness of the connection between endophytes and analyzed the communities as the core hub, which provided the possibility for further selection of the core species. A modularity index above 0.4 suggests that the network has a modular structure [50]. Final visualization was performed using Gephi software v.0.9.2 [51]. To summarize the composition of the major microbiome, we used the *filter* function in the R package *tidyverse* (v.1.3.1) [52] to filter species with a percentage of more than 1% in the bacteria sample at the genus level, and a percentage of more than 10% in the fungi sample at the genus and family level. Subsequently, a community bar plot was generated for each dataset. We used the flower plot implemented in E-venn (http://www.ehbio.com/test/venn, accessed on 13 March 2023 ) to display the common and unique taxa between different samples or sections [53]. The clustering of microorganisms in *Cycas* coralloid roots were described in order to establish the relationships between the core taxa and their proportions among the individual sections or species. All of the bacterial community and fungal community with less than 2% abundance were filtered out to perform phylogeny. The maximum likelihood method was used to calculate the branch distance between the microbiome, and the resulting phylogenetic trees were displayed using the Interactive Tree Of Life (iTOL) website (https://itol.embl.de, accessed on 22 April 2023) with a bubble chart of the proportion of the community in each sample [54]. Candidate-core taxa were defined on the basis that they were in ≥50% of replicates within one or more samples and at a mean relative abundance of ≥0.5% [55]. Here, we consider a taxon as a candidate-core species if they appeared in all sections, and showed a mean relative abundance ≥10% in the fungal community, or ≥1% in the bacterial community. In order to further investigate the microbial communities in coralloid roots and confirm the functional importance of core microbiome, FUNGuild (v1.0) software was used to predict the functions of the fungal community [56], while STAMP software (v2.1.3) was used to generate functional heatmaps [57]. The heatmap of the top 30 functions based on their abundances; species corresponding to functions with abundances above 20% were selected for subsequent analysis, the heatmap of the top 30 of these selected species was drawn.

## 3. Results

### 3.1. Community Structure and Diversity in Coralloid Roots

After quality filtering and denoising the raw sequence, a total of 3,059,752 high-quality reads remained. Out of these, a total of 769,412 reads were mapped to the SILVA 138 database and 2,290,340 reads were mapped to the unite8.0 database (Appendix A). Rarefaction curves indicated that the number of observed ASVs for each individual sample approached saturation, suggesting that the data sequencing was sufficient (Appendix A). The Good’s coverage of all samples was close to 1 (Figure 3a,e, Appendix A), indicating that the sequencing depth was adequate for describing the composition of fungal and bacterial communities. Upon lowering the taxonomic rank, there was a gradual decrease in the proportion of annotated and aligned sequences in both databases. At the species level, the annotation proportion for fungal and bacterial communities were found to be 2.55% and 33.06%, respectively (Appendix A). Nevertheless, at least half of the reads from both fungal and bacterial communities can be assigned at the genus level.

The bacterial Shannon diversity was highest in the IS samples and lowest in the AS, while no significant difference was found among the bacterial community (Figure 3b, Appendix A). Opposite to the bacterial community, the Shannon diversity index of the AS section was found to be the highest for the fungal community, while the IS section had the lowest diversity (Figure 3f, Appendix A). Generally, the Shannon diversity level was higher in the fungal community than in the bacterial community. The Shannon diversity in the fungal community was not statistically significant between the four sections, although there were some differences (Figure 3f). The number of ASVs (Chao1 index) of the bacterial community was found to be the highest in the IS section, followed by PZH and SS, and the AS section had the lowest number of bacterial ASVs (Figure 3c, Appendix A). Interestingly, the observed trend in fungal ASV numbers in the four sections was also opposite to that of the bacterial community (Figure 3g, Appendix A), and the number of fungal ASVs was also found to be much higher than that of the bacterial community.

The rarefied ASV results revealed that the bacterial community was mainly composed of *Nostocaceae*, with high abundance (up to 97%) but low diversity. The PCoA results based on Bray–Curtis dissimilarities between sections showed that at the genus level of the bacterial community, PCoA1 explained 73.6% of the total variation in the separation of the microbiome (Figure 3d), and suggested insignificant differences between different sections (R^2^ = 0.04, *p*= 0.914, dispersion *p* = 0.97) (Table 2, Appendix A). The bacterial community also showed no significant differences (*p* > 0.05 and dispersions *p* > 0.05) among the sections, suggesting a conserved adaptation of *Cyanobacteria* in coralloid roots.

The fungal community’s composition at the genus level was significantly shaped by sections (Figure 3h), with sections accounting for 20.2% of the total variation (*p*= 0.006) (Table 3). The dispersion value was also significant in the test (*p* = 0.008) (Appendix A). To explain the dispersion of data and verify the reliability of the analysis, we conducted separated PCoA analyses using samples from every species of the sect. *Stangerioides* with three other sections, respectively. The fungal community remained significant among the sections (*p* < 0.05 and dispersions *p* > 0.05) (Appendix A). Pairwise Adonis tests between the fungal community composition of samples from different sections identified ten pairs of significantly different fungal genus groups between SS and other sections (IS, AS, PZH) at the genus level (Appendix A).

The overall bacterial community network consisted of 121 nodes and 8006 edges with an average path length of 3.093 edges, a clustering coefficient of 0.721 and a modularity index of 0.448, suggesting a modular structure (Table 4). Overall, 99.96% of the bacterial community were positively correlated, while only 0.04% were negatively correlated. Notably, *Nostocaceae* were negatively correlated with several other families, including *Pseudonocardiaceae* (−0.74297), *Rhizobiaceae* (−0.67475), *Xanthobacteraceae* (−0.63742), *Comamonadaceae* (−0.81176), and *Rhodanobacteraceae* (−0.61621). The bacterial community diagram was mainly divided into seven modules (Figure 3i), where the top three modules demonstrated closer connections and more complex network relationships among species within them. Hub species were concentrated in modules 1 and 3 of the bacterial community network, and these hub species mainly belong to the phylum *Proteobacteria*. *Nostocaceae*, the most abundant family, belonged to module 5, accounting for only 0.83% of the total module abundance.

The fungal community had 732 nodes and 7982 edges with an average path length of 4.067 edges, a clustering coefficient of 0.59, and a modularity index of 0.705 (Table 4), and were all positively correlated, mainly consisting of Ascomycota (58.60%) and Basidiomycota (38.67%). This fungal community can be divided into 16 modules (Figure 3k), and can be further divided into 9 modules after merging the modules with an abundance below 5% into a category of ‘others’. These nine modules were then used to investigate the interaction between fungal community. Modules 2, 11, and 0 showed high levels in abundance (10.93% to 18.44%), hub-species (up to 0.14), and degree (from 50 to 76), with their microbiome closely linking to all the samples together at the genus level. However, species with high clustering (0.776) and hub-species (hub up to 0.1) were mainly found in module 0. Based on the observation of the entire fungal network, Ascomycota (66.7%) and Basidiomycota (31.6%) are the dominant phyla (Figure 3l). This observation was consistent with the statistics between tightly connected modules, where the hub-species were also mainly composed of Ascomycota and Basidiomycota.

### 3.2. Core Members of the Bacterial and Fungal Communities in Cycas Sections

The number of unique members of the fungal microbial community was higher than that of the bacterial community (Figure 4a,b, Appendix A). For the bacterial community, all *Cycas* sections shared *Desmonostoc*_PCC-7422 at the genus level, and *Nostocaceae* at the family level. The *Comamonadaceae* and *Pseudonocardiaceae* at the family level were shared in the SS and IS sections. A total of 23 genera from different families were shared in the fungal community of all sections, from which 17 genera can be identified (Figure 4c,d, Appendix A). Overall, 78.94% of these shared genera belonged to Ascomycota and 15.79% belonged to Basidiomycota.

For the composition of the endophytic bacterial community, three genera, i.e., *Desmonostoc*_PCC-7422, *Nostoc*_PCC-73102, and *unclassified_f__Nostocaceae* from *Nostocaceae*, accounted for more than 97% of the average abundance (Figure 4e). The composition of the fungal community varied slightly in different sections. *Exophiala*, *Diaporthe*, *Leptobacillium*, *Alternaria*, *Fusarium*, and *Paraboeremia* of Ascomycota were both present in all sections and had a high relative abundance (>10% in all samples, Figure 4f). At the family level, the common families included Nectriaceae, Herpotrichiellaceae, Cordycipitaceae, Helotiaceae, Diaporthaceae, Didymellaceae, Clavicipitaceae and Pleosporaceae (>10% in all samples, Figure 4d,g, Appendix A).

Based on the clustering of endophytic bacteria shared in the sections, the genus from *Nostocaceae* formed a monophyletic clade, indicating that they had a close relationship (Figure 5a). We further performed a cluster analysis of *Nostocaceae* to show the distribution and relationship of the common genera *Desmonostoc*_PCC-7422 with other *Cyanobacteria Nostoc*_PCC-73102 and *unclassified_f_Nostocaceae* among the samples (Figure 5c). The occurrence and abundance of ASVs in *Nostoc*_PCC-73102 were found to be significantly lower compared to other samples. Notably, ASV1250 derived from the genus *Desmonostoc*_PCC-7422 accounted for the highest proportion in most of the samples (except for CY sample, Figure 5c).

Among the fungal community, a total of 68 genera were filtered out to perform phylogeny. Ascomycota accounted for a high proportion and degree of occurrence in each sample group (Figure 5b). Out of the 23 fungal genera that appeared in all sections, 16 belonged to the phylum Ascomycota. However, these shared genera did not cluster in the same clade, nor they did show a similar pattern in abundance across closely related species or sections. *Exophiala* was found in all the samples and had the highest abundance (except for ST, which had a relatively low abundance), indicating its significance as an important genus within the *Cycas* fungal community. In addition to *Exophiala* and *Leptobacillium* in the ST sample, *Fusarium* in the BC and DB samples and *Paraboeremia* in the GZ sample also showed high abundance. Although all four sections have their “candidate-core” taxa, the core genera that each section/species enriched were different.

## 4. Discussion

### 4.1. A Dynamic Equilibrium Process of Endophytes in Coralloid Roots

Cycads, as ancient plants, exhibit evolutionary conservatism and are the sole gymnosperms discovered thus far that can coexist with *Cyanobacteria*. Coralloid roots, a specialized type of root, harbor a diverse array of endophytes [3,19,29,31,33], among which *Cyanobacteria* attracts wider attention because it can establish stable colonization on coralloid roots and form symbiotic relationships that facilitate cycad growth through nitrogen fixation [58]. This symbiotic process is believed to be dynamic, with the plant-associated microbiome being recruited and assembled during the life cycle of its plant host. Specifically, the microbiome displays high dynamism during the early vegetative phase, gradually converging during vegetative growth, and stabilizing during the reproductive phase [59,60,61]. Additionally, other studies have found that plants may choose and acquire a functional microbiome at a certain stage, and genes related to nitrogen assimilation and carbon degradation become enriched at a late stage [59,62,63].

In this study, we demonstrate that endophytes were also at a convergent and stable stage in the mature coralloid roots of different species cycads, maintaining a dynamic equilibrium. Specifically, the TQ group exhibited highest diversity of bacteria families, but showed the lowest abundance in the fungal community (Figure 4b,d,i). This pattern was also observed at the genus level for both bacteria and fungi in all samples (Figure 4a,c,h). The opposite trend of diversity between endophytic bacteria and fungi suggested the possibility of niche competition and that the population changes of endophytic were dynamic. While it is possible that coralloid roots recruit a specific microbiome during the mature stage to optimize the growth of cycad plants, this trade-off between bacterial and fungal communities in coralloid roots might support the idea that the diversity and abundance of endophytic microbiomes are not static, but rather undergo a process of stable dynamical equilibrium.

The reverse stability and dynamic proportion between bacterial and fungal communities appear to be driven by changes in root metabolic status during the maize lifecycle [64,65], and negative correlations between bacteria and fungi play a role in maintaining plant health [66]. Synergistic interactions among diverse and functionally complex symbiotic microorganisms within the roots can modulate gene expression involved in root development and nutrient transport [67]. Interactions between microbiome members also contribute to the establishment, stability, and resilience of host-associated microbial communities [68]. Consequently, it might be that the different functional requirements of different sections of *Cycas* lead to the dynamic changes in microbiome in their coralloid roots. From an evolutionary standpoint, the ability of a symbiont to coordinate the flow of resources based on the inputs of its partners determines the sustainability of a mutually beneficial symbiosis. Therefore, it was extremely important to have a stable, healthy, and dynamic microbiome.

In addition, we also found that *Cycas* section/species with close relationships had dynamic relationships similar to the microbiome. In previous phylogenetic analyses on *Cycas* [17,69], the sect. *Panzhihuaenses* had the closest relationship with sect. *Asiorientales*. A similar dynamic trend of endophytic proportion could be observed between the two sections compared to others. Within the sect. *Stangerioides*, *C. chenii* (CS) and *C. guizhouensis* (GZ) are sister species, and similar patterns of microbiome dynamics can be revealed. It should not be overlooked that, due to the larger number of samples involved, the changes in the microbiome of *Stangerioides* were more variable than those of the other sections. Nevertheless, the dynamic shifts in microbiomes across each section or species indicate that these changes and the stable presence of the core microbiome may also be a response to various signals, coordinating the assembly of microbes within the coralloid roots and maintaining the healthy growth of cycads.

### 4.2. Core Species and Their Potential Roles

The plant core microbiome consists of microbial communities that are persistent and universal in almost all communities. This core microbiome contains key microbial taxa that harbor genes essential for host fitness [14]. In this study, we observed that the endophytic bacteria were predominantly *Nostocaceae*, with a notable abundance of *Desmonostoc*, while the fungal community was mainly composed of Ascomycota. Although *Proteobacteria* exhibited the highest number of genera within the bacterial community (34.71%, Figure 3j), the core bacteria in coralloid roots are mainly composed of *Cyanobacteria*, which only represent 2.48% of all genera (Figure 3j). At the family level, *Nostocaceae* displayed low diversity across all samples but had a particularly high abundance, and it played a key connecting role in the bacterial co-occurrence network which might form a stable symbiotic relationship with the coralloid roots of *Cycas*. *Pseudonocardiaceae* and *Comamonadaceae* were present in the sect. *Stangerioides* and *Indosinenses* but showed a negative correlation with *Nostocaceae* in the co-occurrence network analysis. Despite the fact that rare microbes may also contribute to host-associated functions [70], the abundance of *Comamonadaceae* was even less than 1% across most of the samples, making it unlikely to be considered a core bacterium. In contrast, the abundance of *Pseudonocardiaceae* was higher than 1% in most of the sections and can be “candidate-core taxon” of endophytic bacterial communities of cycad coralloid roots after *Nostocaceae*.

*Nostoc* was the main *Cyanobacteria* that had previously been found in the coralloid roots of cycads [27]. However, our findings demonstrate that *Desmonostoc*, rather than the endophyte *Nostoc*, was more prevalent in the coralloid roots across all *Cycas* sections. While previous studies have reported the presence of *Desmonostoc* in coralloid roots [31], the importance of this genus in coralloid root endophytes might have been overlooked. *Desmonostoc* can usually be found in moist or wet meadow, field, and forest soils, but is rare or absent in desert areas [71]. The suitable environment for *Desmonostoc* is aligned with the optimal habitat of cycads, which are predominantly found in tropical and subtropical regions. *Desmonostoc* can grow and maintain photosynthesis at two-times higher salinity than *Nostoc* [72], showing strong antifungal activity [73], and can secrete a secondary metabolite to defend against plant pathogens and promote plant growth [74]. The various known properties of *Desmonostoc* might thus offer potential to assist cycads in resisting harsh environments, despite the fact that the additional roles it plays within the coralloid roots remain to be further investigated.

In addition, the complexity of endophytic fungi was significantly higher than that of endophytic bacteria. There were 23 shared fungal genera in all samples, with 20 genera identified below the family level, indicating a relatively high abundance of shared genera as the core species. *Exophiala*, as a highly abundant shared genus across all sections, may have played a crucial role in the adaptation of cycads and their survival from harsh environments. The functional profile of coralloid roots’ microbiome showed remarkable similarity among different *Cycas* species or sections (Figure 6a). The heatmap analysis (Figure 6b) revealed that *Exophiala* and Didymellaceae remained dominant among species with high functional abundance. This finding aligns with the selection of core species based on their composition within the fungal community, where their abundance accounted for more than 10% in samples (Figure 4f). Functional predictions further confirmed that *Exophiala* and Didymellaceae are core species playing an important role in coralloid roots (Figure 6), thus solidifying their status as essential members of the coralloid root microbiome. A recent study demonstrated that symbiotic *Exophiala* can promote maize growth by regulating the expression of genes involved in plant hormone signaling and polar transport in maize roots, which affected abscisic acid (ABA) and indole-3-acetic acid (IAA) contents [75]. The abundance of *Exophiala* from the biological soil crusts was found to have a positive correlation with microbial biomass nitrogen (MBN), electrolytic conductivity (EC), nitrate (NO_3_^−^), suggesting its potential for nitrogen fixation [76]. Additionally, *Exophiala* has demonstrated the ability to resist various forms of stress [77], improve plant tolerance to adverse environment and promote host plant growth [78,79]. Meanwhile, according to our results, Nectriaceae had also been shown to be a core fungal family in coralloid roots of all *Cycas* sections, suggesting a universality of the family in *Cycas*. This suggestion was in congruence with a previous finding that Nectriaceae was extremely abundant in different cycad tissues, such as seeds, ovules, pollens, and roots from *C. panzhihuaensis* [3]. Thus, it is likely that *Exophiala* and Nectriaceae are the most important endophyte fungi for *Cycas* adaptation to the physical and chemical properties of climate and soil within ex situ conserved area. This raises further attention that should be paid to the interaction of these two groups with *Cycas*.

The fungal networks exhibited greater complexity compared to the bacterial networks, and the interconnections between microbial communities were also more tightly integrated within the fungal community (Figure 3i–l). This is likely because there is more species diversity shared across samples in fungal than bacterial communities. For instance, in the fungal community, in addition to Nectriaceae, there were shared families such as Herpotrichiellaceae, Cordycipitaceae, Helotiaceae, Diaporthaceae, Didymellaceae, Clavicipitaceae, and Pleosporaceae. At the genus level, apart from *Exophiala*, the fungal core taxa also included *Diaporthe*, *Leptobacillium*, *Alternaria*, *Fusarium* and *Paraboeremia*. However, only one shared species found in the bacterial community was *Desmonostoc* of the family *Nostocaceae*. Furthermore, the numbers of core fungi in the fungal community were significantly higher than that observed in the bacterial community, suggesting an overall more dominant position of fungi during the microbial assembling process. In spite of this, fungi and bacteria were both potentially important partners for *Cycas* in growth and reproduction, as the enrichment and stable presence of the core endophytes in the host are mainly attributed to the function of the microbiome rather than classification. More and more recent studies have confirmed that the plant-associated core microbiome provide benefits to plants through various direct or indirect mechanisms [14]. For example, core endophytes (both fungi and bacteria) have played a role in transmitting signals to control nutrient uptake by plants [65], and can inhibit root diseases by producing antibiotics and help plants to adapt better to the environment [80,81]. Likewise, the roles and functions of microbiomes in the coralloid roots shared between different taxa may be the basis of their symbiosis with the host which has been captured and stabilized by them.

## 5. Conclusions

In this study, we discovered extremely diverse and intricately structured microbial communities in coralloid roots of various *Cycas* species from the same growth environment. Nevertheless, there are identifiable core communities among all sections of *Cycas*. Generally, *Cyanobacteria* and Ascomycota dominate the endophytic microbiome. The genera *Desmonostoc* and *Exophiala* exhibited relatively higher abundance and play key roles at the genus level in bacterial and fungal communities, respectively. Furthermore, we confirmed the universality of Nectriaceae, a family found throughout different compartments in *C. panzhihuaensis* [3], as well as coralloid roots from different sections of *Cycas*; this finding warrants further attention in future microbial studies on *Cycas.* The establishment of root microbiota distinct from those found in the soil is a dynamic process that initially resembles their soil origins but becomes more plant-specific as the plant grows [63,82]. This study enhances our understanding of symbiosis between the microbiome and cycad coralloid roots in an ex situ environment. Future studies utilizing comprehensive metagenomic sequencing and including more *Cycas* species from different wild habitats will gain us a better understanding of the functions of different microbial communities in the coralloid roots, and provide insights into the adaptation of cycads to arid and barren habitats.

## Figures and Tables

**Figure 1 microorganisms-11-02144-f001:**
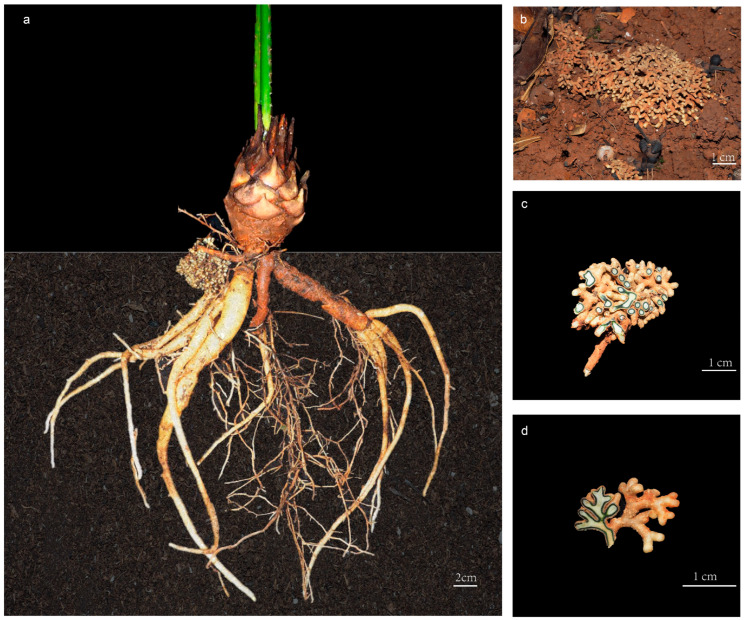
The coralloid roots of *Cycas* under ex situ protection in Kunming Botanical Garden, Yunnan, China: (**a**) morphology of the whole root of *Cycas*; (**b**) growing status of cycad coralloid root; (**c**) cross sectional illustration of coralloid root; (**d**) longitudinal illustration of coralloid root.

**Figure 2 microorganisms-11-02144-f002:**
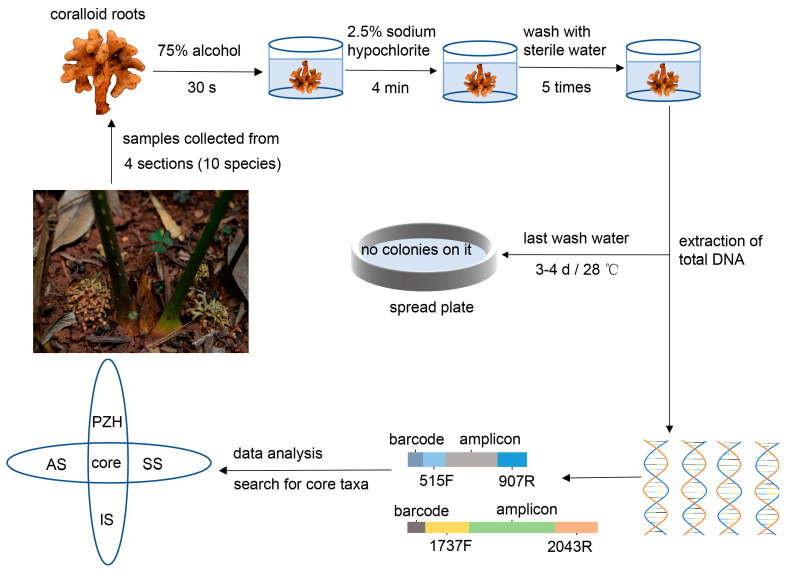
The comprehensive experimental design.

**Figure 3 microorganisms-11-02144-f003:**
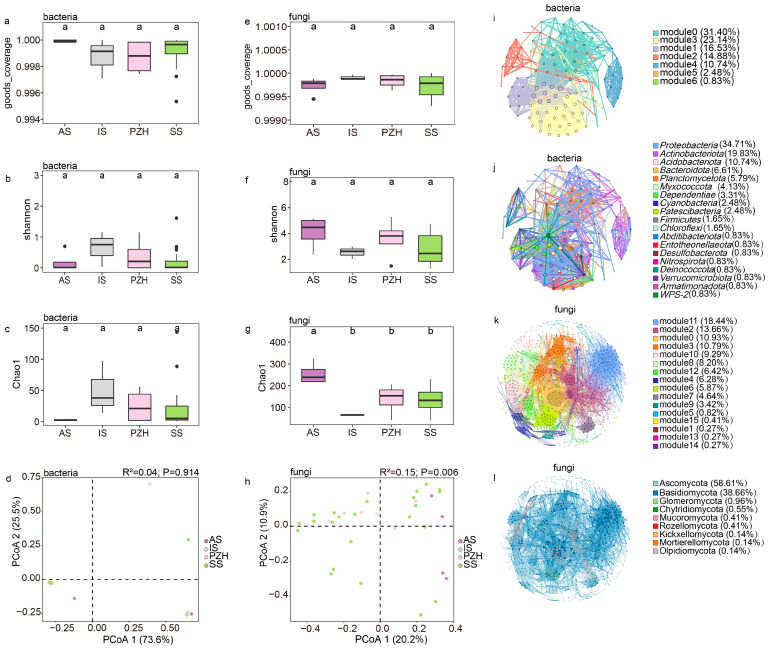
The structure of endophytic bacteria and fungal communities in mature coralloid roots of *Cycas*: bacterial (**a**–**c**) and fungal (**e**–**g**) α-diversity measured across different sections at the genus level; different letters were used to indicate the significance of the difference (*alpha* = 0.05). (**d**) principal coordinate analysis of the bacterial community at the genus level; (**h**) principal coordinate analysis of the fungal community at the genus level; (**i**) modular co-occurrence of endophytic bacteria at the family level; (**j**) co-occurrence network of phyla in the endophytic bacteria community at the family level; (**k**) modular co-occurrence of endophytic fungi at the genus level; (**l**) co-occurrence network of phyla in the endophytic fungi community at the genus level. Larger nodes indicate more connections to other nodes. Wider line segments indicate stronger correlations between connected nodes.

**Figure 4 microorganisms-11-02144-f004:**
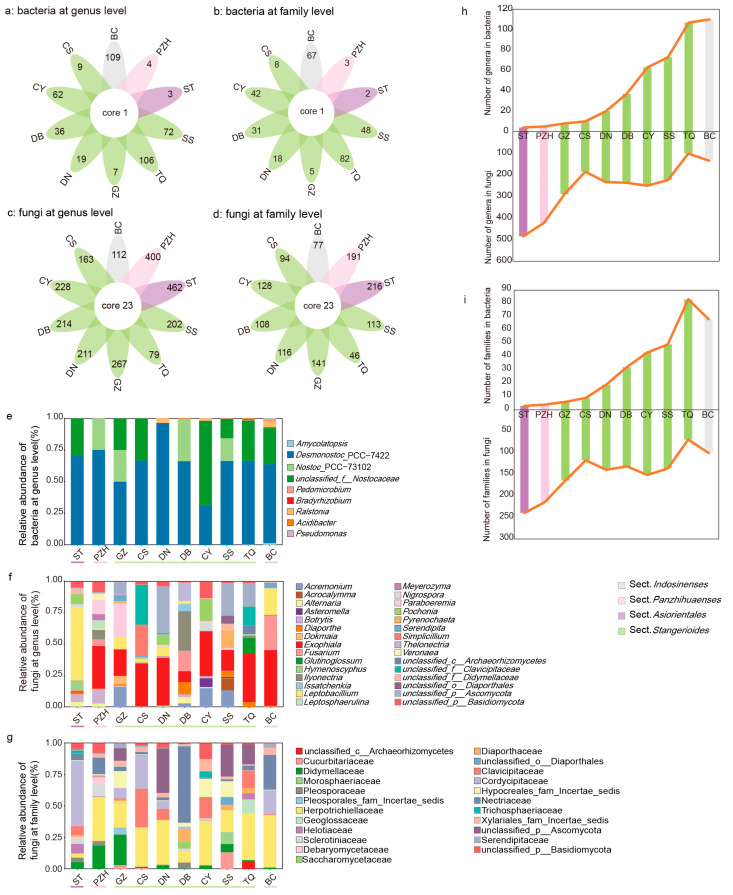
Communities of bacteria and fungi in four sections of *Cycas*: (**a**–**d**) the comparison of endophytic communities across different sections of *Cycas* based on 16s rRNA (**a**,**b**) and ITS sequences (**c**,**d**) at the genus or family level; (**e**–**g**) taxonomical composition of coralloid root endophytic bacteria at the genus level (**e**) and fungi (**f**,**g**) at the genus or family level in different *Cycas* species; (**h**,**i**) microbiome dynamic balance of coralloid root endophytes at the genus (**h**) and family (**i**) level. Taxa are labelled with an identifiable taxonomic level if they failed to be assigned at the genus or family level.

**Figure 5 microorganisms-11-02144-f005:**
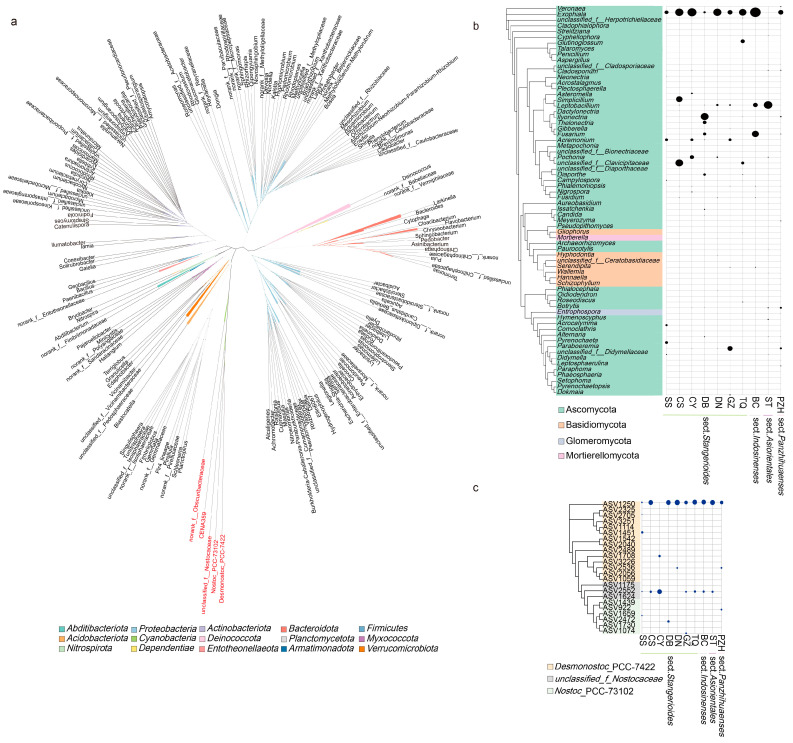
Phylogenetic tree of abundant bacterial (left panel) and fungal (right panel) species colonizing cycad oralloid roots: (**a**) unrooted tree of all endophytic bacteria; the different genera of *Nostocaceae* highlighted in red; (**b**) clustering of fungal genera with abundance higher than 2%, where the abundance of genera were shown in the right panel; (**c**) clustering of *Nostocaceae* ASVs with the abundance of ASVs displayed in the right panel. For (**b**,**c**), family-level classification was used if the genus was unknown. Taxa are labelled with an identifiable taxonomic level if they could not be assigned at the genus or family level. Relative abundances were aggregated at the family (or genus) level and are depicted with circles.

**Figure 6 microorganisms-11-02144-f006:**
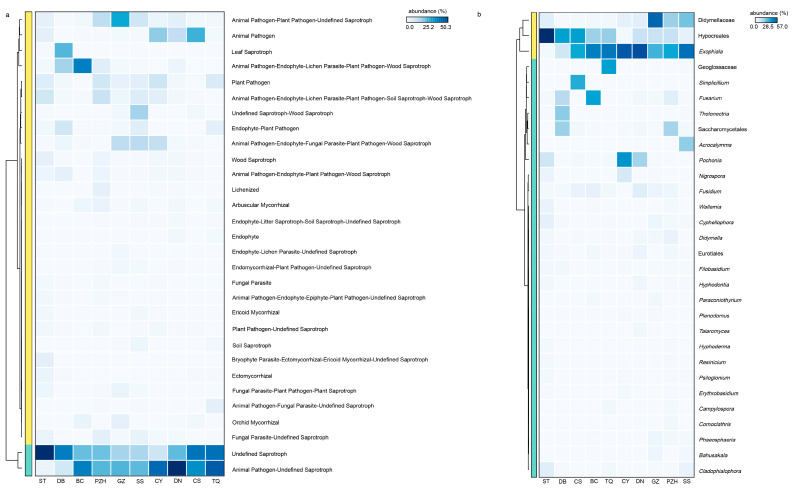
The functional prediction of the fungal community and its core microbiome with high-abundance functional components. (**a**) Functional heatmap displaying the top 30 abundantly functional components in the fungal community; (**b**) species heatmap illustrating the top 30 abundant species with high functionality (proportion of abundance > 20%). Family-level or order-level classification was utilized when the genus was unknown.

**Table 1 microorganisms-11-02144-t001:** List of root samples and their host species used in this study.

Coralloid Root	Host Species	Taxonomic Status
CS	*Cycas chenii* X. Gong & Wei Zhou	sect. *Stangerioides*
CY	*Cycas bifida* (Dyer) K. D. Hill	sect. *Stangerioides*
DB	*Cycas debaoensis* Y. C. Zhong & C. J. Chen	sect. *Stangerioides*
DN	*Cycas diannanensis* Z. T. Guan & G. D. Tao	sect. *Stangerioides*
GZ	*Cycas guizhouensis* K. M. Lan & R. F. Zou	sect. *Stangerioides*
SS	*Cycas sexseminifera* F. N. Wei	sect. *Stangerioides*
TQ	*Cycas tanqingii* D. Y. Wang	sect. *Stangerioides*
BC	*Cycas pectinata* Buchanan-Hamilton	sect. *Indosinenses*
PZH	*Cycas panzhihuaensis* L. Zhou & S. Y. Yang	sect. *Panzhihuaenses*
ST	*Cycas revoluta* Thunb.	sect. *Asiorientales*

**Table 2 microorganisms-11-02144-t002:** PERMANOVA of bacterial community at genus level for four *Cycas* sections.

	Df	Sum of Sqs	R^2^	F	Pr (>F)
group	3	0.285825619	0.035485913	0.3679149	0.914
Residual	30	7.768796599	0.964514087		
Total	33	8.054622218	1		

**Table 3 microorganisms-11-02144-t003:** PERMANOVA of fungal community at genus level for four *Cycas* sections.

	Df	Sum of Sqs	R^2^	F	Pr (>F)
group	3	1.665261	0.147219	1.726344	0.006
Residual	30	9.64617	0.852781		
Total	33	11.31143	1		

**Table 4 microorganisms-11-02144-t004:** Main estimated parameters of two co-occurrence networks of the microbiome in the coralloid roots.

Parameter	Bacteria	Fungi
Number of nodes	121	739
Number of edges	1160	8006
Average clustering coefficient	0.721	0.59
Average path length	3.093	4.067
Graph density	0.16	0.029
Modularity	0.448	0.705
Network diameter	7	12
Average weighted degree	30.281	33.063
Average degree	19.174	21.667

## Data Availability

Sequencing data are available in the NCBI Sequence Read Archive (SRA) with the accession BioProject ID: PRJNA1007376.

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
