# Peer review of "Core Microbiome and Microbial Community Structure in Coralloid Roots of Cycas in Ex Situ Collection of Kunming Botanical Garden in China"

_microorganisms, 2023, doi:10.3390/microorganisms11092144_

Round 1

Reviewer 1 Report

The manuscript by Wang et all. presents the analysis of bacterial and fungal communities in coralloid roots of Cycas species in China. These results are of potential interest to a broad audience, specifically those involved in environmental science and core microbiome of plant roots analysis. The paper is generally well-structured and but it needs a lot of improvements. First of all, the English usage in the paper needs improvement. I strongly suggest getting editing help from someone with full professional proficiency in English. Moreover:
1.       Italicize the Latin bacteria names regardless of the taxonomic level
2. Figure 1 – contains the methodology information. I suggest to separate some information ((e) – part) and moving it into the materials and methods section
3. Line 109-112 “Cycas pectinata (BC) is the most widely distributed species in sect. Indosinenses Schuster (IS), Cycas panzhihuaensis (PZH) is the only species in sect. Panzhihuaenses (D. Yue Wang) K.D. Hill (PZH), Cycas revoluta (ST) is the only species in sect. Asiorientales Schuster (AS) [36], and other species are all from sect. Stangerioides Smitinand (SS) covering most habitats of this section [37]” – this description is chaotic and incomprehensible to me. Please try to improve it.
4. Line 130-131 – the authors provide only one thermocycling condition. Is it optimal both for 16SrRNA and ITS?
5. Please provide more information about sequencing (e.g. what kits you used, how long the reads are)
6. Figure 2 and Figure 4 are unreadable. Please increase the font
7. Conclusion section – provide more information about the future directions of this research.

Moderate editing of English language required

Author Response

Response to Reviewer 1

Point 1: the English usage in the paper needs improvement.

Response 1: The English has been refined and rechecked throughout the text.

Point 2: Italicize the Latin bacteria names regardless of the taxonomic level.

Response 2: Necessary corrections were made to italicize all the bacteria Latin names. Additionally, all fungal genera in Figure 5b and Figure 6b were appropriately formatted in italics.

Point 3: Figure 1 – contains the methodology information. I suggest to separate some information ((e) – part) and moving it into the materials and methods section.

Response 3: The (e)-part in Figure 1 has been extracted and presented as a separate entity in Figure 2, which located in materials and methods section in the manuscript. We replaced Figure 1 as a more visually distinct and comprehensive of coralloid roots. The initial labels of Figure 2-4 were sequentially updated to the new figures: Figure 3-5 respectively. These modifications have also been incorporated into the revised manuscript.

Point 4: Line 109-112 “Cycas pectinata (BC) is the most widely distributed species in sect. Indosinenses Schuster (IS), Cycas panzhihuaensis (PZH) is the only species in sect. Panzhihuaenses (D. Yue Wang) K.D. Hill (PZH), Cycas revoluta (ST) is the only species in sect. Asiorientales Schuster (AS) [36], and other species are all from sect. Stangerioides Smitinand (SS) covering most habitats of this section [37]” – this description is chaotic and incomprehensible to me.

Response 4: In the revised manuscript: we offered more detailed reason of the sampling strategy: ‘Currently, the distribution of Cycas in China is primarily classified into four distinct sections. To enhance our comprehensive understanding of the existing endophytes in Cycas coralloid roots within China, core microbiome suitable for most Cycas sections were selected whenever possible, and representative species from each of the four sections were chosen. These selected Cycas species are all located within the ex-situ conservation area of Kunming Botanical Garden. The rationale behind selecting these representative species from each section is as follows: Cycas panzhihuaensis (PZH) is the only species in sect. Panzhihuaenses (D. Yue Wang) K.D. Hill (PZH), Cycas revoluta (ST) is the only species in sect. Asiorientales Schuster (AS), Cycas pectinata (BC) is the most widely distributed species in sect. Indosinenses Schuster (IS), and other species from the sect. Stangerioides Smitinand (SS) cover most of the habitats of this section. The design and methodology of the experiment are illustrated in Figure 2.’

We hope this revision is more clear and easy to understand.

Point 5: Line 130-131 – the authors provide only one thermocycling condition. Is it optimal both for 16SrRNA and ITS?

Response 5: Yes, according to our laboratory experience, the thermocycling conditions provided in the manuscript for PCR amplification of Cycas coralloid roots using performed quite well for both 16SrRNA and ITS primers, allowing for maximum amplification of the targeted sequences when using different cycles (27 and 35).

Point 6: Please provide more information about sequencing (e.g. what kits you used, how long the reads are).

Response 6: Information regarding the kits utilized and read length have been incorporated into the manuscript.

Point 7: Figure 2 and Figure 4 are unreadable. Please increase the font.

Response 7: The font size of Figure 3 (former Figure 2) and Figure 5 (former Figure 4) has been enlarged in the revised manuscript.

Point 8: Conclusion section – provide more information about the future directions of this research.

Response 8: Thank you. In the revised manuscript, we strengthened significance of this study and highlighted future directions at the end of the conclusion section.

Reviewer 2 Report

The study of the Core microbiome and microbial community structure in coralloid roots of Cycas in China focuses on the analysis of coralloid roots in the collection (ex-situ conservation) of the Kunming Botanical Garden.

As a fairly comprehensive study on the genus Cycas it certainly provides interesting information.

Reading the paper, I have the impression that for the effort made in terms of analysis, bioinformatics, the conclusions are rather obvious and well known from the literature in the field. Indeed, the effort made in terms of networking seems commendable, but frankly the conclusions are rather predictable.

In general, when the author wants to investigate a core microbiome of such an important genus (Cycas) in the evolution of seed plants, he cannot ignore the comparison between the collection and the species in the wild. I repeat, if we want to analyse the core microbiome of Cycas, we must bear in mind the environmental adaptations and edaphic conditions found in nature. For it is these conditions that select the key species in a core microbiome.

Furthermore, the metagenomic analysis carried out is very quantitative and not very qualitative, I would add the perspectives of functional metagenomics.

I therefore suggest revising the title by changing it to: Core microbiome and microbial community structure in coralloid roots of Cycas in ex-situ collection of Kunming Botanical Garden in China.

Add more detailed functional information on the different functions of the microorganisms found and possibly identify key species beyond those already generically mentioned.

A moderate revision of English is required

Author Response

Response to Reviewer 2

Point 1: In general, when the author wants to investigate a core microbiome of such an important genus (Cycas) in the evolution of seed plants, he cannot ignore the comparison between the collection and the species in the wild. I repeat, if we want to analyse the core microbiome of Cycas, we must bear in mind the environmental adaptations and edaphic conditions found in nature. For it is these conditions that select the key species in a core microbiome.

Response 1: We fully agree that both the environmental adaptation and soil conditions are crucial and significant factors that influence the core community of coralloid roots. However, environmental conditions vary greatly among different wild Cycas species (populations), thus making it difficult to distinguish the environmental roles in recruiting core microbiome species. Environmental adaptation of core microbiome is also an important topic but is beyond the scope of this study, which focuses on the impact of species factors on the endophytes of different Cycas coralloid roots within the same ex-situ conservation environment. Our main objective is to identify shared core microbiome among different Cycas species or taxonomic groups, and to predict their respective functions.

However, it will be very interesting to include Cycas coralloid root samples from different natural populations in future studies to examine the influence of environmental or soil factors in shaping the core microbiome.

Point 2: Furthermore, the metagenomic analysis carried out is very quantitative and not very qualitative, I would add the perspectives of functional metagenomics.

Response 2: In this study, amplicon sequencing was employed to investigate the diversity and abundance of endophytes in mature coralloid roots and elucidate the role of microorganisms by function prediction (Figure 6). Compared to metagenomic sequencing, our amplicon sequencing indeed bears with significant disadvantages in function prediction. In the subsequent studies, we will apply metagenomics sequencing technology to predict microorganisms’ functions in Cycas coralloid roots.

Point 3: I therefore suggest revising the title by changing it to: Core microbiome and microbial community structure in coralloid roots of Cycas in ex-situ collection of Kunming Botanical Garden in China.

Response 3: Thanks for this advice, the title has been revised to: ‘Core microbiome and microbial community structure in coralloid roots of Cycas in ex-situ collection of Kunming Botanical Garden in China’ as suggested.

Point 4: Add more detailed functional information on the different functions of the microorganisms found and possibly identify key species beyond those already generically mentioned.

Response 4: Thanks for this comment. In the revised manuscript, we added more detailed functional information of the microorganisms and identified extra key species. For example, we added extra analysis for predicting fungal function The results indicated no significant differences in functionality among the samples. We also selected the first 30 functional abundances to generate functional heatmaps (Figure 6a). Furthermore, by examining species exhibiting a high functional abundance (Figure 6b), we provide additional evidence for the importance of Exophiala identified in this study, for which their involvement in signal transmission, stress resistance, and even nitrogen fixation were demonstrated in recent studies. We speculated that Exophiala may play a crucial role in promoting Cycas coralloid root health, despite further investigation are needed.

Round 2

Reviewer 1 Report

Authors corrected the manuscript according to my suggestions. I am satisfied and in my opinion revised version is acceptable.